# Graphene-Based Coating to Mitigate Biofilm Development in Marine Environments

**DOI:** 10.3390/nano13030381

**Published:** 2023-01-18

**Authors:** Francisca Sousa-Cardoso, Rita Teixeira-Santos, Ana Francisca Campos, Marta Lima, Luciana C. Gomes, Olívia S. G. P. Soares, Filipe J. Mergulhão

**Affiliations:** 1LEPABE—Laboratory for Process Engineering, Environment, Biotechnology and Energy, Faculty of Engineering, University of Porto, Rua Dr. Roberto Frias, 4200-465 Porto, Portugal; 2ALiCE—Associate Laboratory in Chemical Engineering, Faculty of Engineering, University of Porto, Rua Dr. Roberto Frias, 4200-465 Porto, Portugal; 3LSRE-LCM—Laboratory of Separation and Reaction Engineering—Laboratory of Catalysis and Materials, Faculty of Engineering, University of Porto, Rua Dr. Roberto Frias, 4200-465 Porto, Portugal

**Keywords:** marine biofouling, antifouling surfaces, graphene, *Cobetia marina*, biofilm formation

## Abstract

Due to its several economic and ecological consequences, biofouling is a widely recognized concern in the marine sector. The search for non-biocide-release antifouling coatings has been on the rise, with carbon-nanocoated surfaces showing promising activity. This work aimed to study the impact of pristine graphene nanoplatelets (GNP) on biofilm development through the representative marine bacteria *Cobetia marina* and to investigate the antibacterial mechanisms of action of this material. For this purpose, a flow cytometric analysis was performed and a GNP/polydimethylsiloxane (PDMS) surface containing 5 wt% GNP (G5/PDMS) was produced, characterized, and assessed regarding its biofilm mitigation potential over 42 days in controlled hydrodynamic conditions that mimic marine environments. Flow cytometry revealed membrane damage, greater metabolic activity, and endogenous reactive oxygen species (ROS) production by *C. marina* when exposed to GNP 5% (*w*/*v*) for 24 h. In addition, *C. marina* biofilms formed on G5/PDMS showed consistently lower cell count and thickness (up to 43% reductions) than PDMS. Biofilm architecture analysis indicated that mature biofilms developed on the graphene-based surface had fewer empty spaces (34% reduction) and reduced biovolume (25% reduction) compared to PDMS. Overall, the GNP-based surface inhibited *C. marina* biofilm development, showing promising potential as a marine antifouling coating.

## 1. Introduction

Marine environments host numerous surfaces, both naturally occurring and manmade, with distinct physical and chemical properties. These submerged surfaces are subject to the rapid accumulation of organisms and organic matter, in a process known as biofouling. Marine biofouling is a widespread concern with severe economic and environmental consequences [1,2]. Its nefarious effects are particularly clear in the naval industry, where the attachment and colonization of ship hulls by fouling organisms contribute to surface deterioration and significantly increase the watercraft’s drag force, leading to higher fuel consumption and greater maintenance costs [3,4]. Furthermore, greater fuel consumption caused by marine biofouling implies increased levels of greenhouse gas emissions [5]. Biofouling on oceangoing vessels can also promote the introduction of invasive exotic species into non-native environments as well as the contamination of aquaculture facilities, therefore harming global biodiversity and raising public health concerns, respectively [6,7,8]. Lastly, the adhesion of fouling organisms on marine surfaces can also interfere with the measurements of underwater sensors, and contribute to the deterioration of submerged infrastructures [9,10,11].

Marine biofouling involves a wide variety of organisms [12,13]. According to their dimensions and level of complexity, fouling organisms can be divided into micro- and macrofoulers. Even though biofouling does not follow a fixed order of events, it is generally considered that surface colonization by microfoulers precedes and promotes attachment by macrofoulers, which highlights biofilm formation as a crucial step in the marine biofouling process [14,15].

Among the wide range of organisms that comprise marine biofilms, *Cobetia marina* (formerly *Deleya marina*) has been extensively used in marine biofouling research as a model marine biofilm-forming bacterium [16]. This Gram-negative, rod-shaped bacterium possesses several features that make it relevant when studying bacteria–surface interactions, such as extracellular polymeric substances (EPS) production and gliding motility, which allow it to promptly colonize surfaces and form stable biofilms [17,18,19]. In fact, *C. marina* is known to produce large quantities of EPS [20,21,22]. This is particularly relevant since it has been demonstrated that one of the most prominent challenges in marine biofouling mitigation is linked to microorganisms that are both capable of initiating biofilm development, as well as ensuring its cohesion, by excreting large amounts of EPS [23]. Thanks to these unique features, *C. marina* can be considered a representative marine microfouler.

Marine biofilm formation is a nanoscale interfacial phenomenon. As such, innovative surface engineering techniques that modify surface attributes at the nanoscale can not only optimize the properties of certain materials [24,25], but also provide them with significant antifouling potential [26,27]. In fact, the incorporation of nanomaterials into marine antifouling paints has been reported to greatly impact a surface’s charge potential, hydrophobicity, and topography, as well as its antibacterial and anticorrosion properties [28,29]. Among these emerging nanoengineered antifouling paints, those containing carbon nanomaterials, such as carbon nanotubes and graphene, have shown promising results [30,31,32].

Graphene is one of the strongest and thinnest materials available [33]. It consists of a single-layer sheet of sp^2^-hybridized carbon atoms with a two-dimensional hexagonal structure. Due to their high specific surface area, electrical conductivity, and thermal stability, graphene-based materials are very appealing for multiple applications [34,35]. Furthermore, graphene is recognized for its antimicrobial and anti-adhesive properties [36,37]. Even though this carbon material’s mechanisms of action are not yet fully understood, it is postulated that the sharp edges of graphene sheets can lead to membrane damage and bacterial cell entrapment. Additionally, graphene is assumed to induce oxidative stress through the formation of reactive oxygen species (ROS), which disrupt microorganisms’ DNA and proteins [38]. As such, the incorporation of graphene in marine paints can not only improve their mechanical strength and durability, but also provide them with enhanced antifouling attributes [32,39].

The main objective of this study was to assess the effect of a graphene-based coating on biofilm development by *C. marina* over a long-term in vitro assay performed under hydrodynamic conditions mimicking a real marine setting. Furthermore, the mechanism of action of pristine graphene nanoplatelets (GNP) was investigated.

## 2. Materials and Methods

### 2.1. Bacterial Strain and Culture Conditions

The *C. marina* (strain DSM 4741/ATCC 25374) was isolated from a coastal sea sample near Woods Hole (Falmouth, MA, USA) [40] and purchased from the Leibniz Institute DSMZ—German Collection of Microorganisms and Cell Cultures (Braunschweig, Germany).

*C. marina* stored in frozen aliquots was streaked onto a Våatanen Nine Salt Solution (VNSS) agar plate, prepared according to Holmström et al. [41], and supplemented with 14 g·L^−1^ agar (VWR International S.A.S., Fontenay-sous-Bois, France). After 24 h of bacterial growth at 25 °C, individual bacterial colonies from the plate were inoculated into 100 mL sterile VNSS medium and incubated overnight in an orbital shaker with a 25 mm diameter, at 25 °C and 120 rpm (Agitorb 200ICP, Norconcessus, Ermesinde, Portugal). The bacterial culture was then centrifuged (Eppendorf Centrifuge 5810R, Eppendorf, Hamburg, Germany) for 10 min at 3100× *g*. The pellet was resuspended in sterile VNSS medium, and the cell suspension was diluted to an optical density at 610 nm (OD_610_) of 0.1 (3.5 × 10^7^ cells·mL^−1^) (V-1200 spectrophotometer, VWR International China Co., Ltd., Shanghai, China). At this OD_610_, bacteria are in the log phase, which the literature suggests to be the growing phase that provides the most reliable results [42].

### 2.2. Determination of GNP Minimal Inhibitory Concentration

Minimal Inhibitory Concentration (MIC) was determined by exposing *C. marina* to GNP (Alfa Aesar, Thermo Fisher Scientific, Erlenbachweg, Germany) for 24 h at 25 °C under static conditions. The tested GNP concentrations ranged between 0.15 and 5% (*w*/*v*). MIC was defined as the lowest concentration that prevented any discernible growth.

### 2.3. Characterization of GNP Mechanisms of Action

#### 2.3.1. Flow Cytometry Analysis

Graphene’s mechanism of action was characterized in *C. marina* using flow cytometry. Briefly, a bacterial suspension containing approximately 3.5 × 10^7^ cells·mL^−1^ was exposed to GNP 5% (*w*/*v*) for 24 h at 25 °C. Afterwards, bacterial cells were harvested by centrifugation at 9715× *g* (Eppendorf 5418, Eppendorf AG, Hamburg, Germany) for 10 min at room temperature and the supernatant was collected for analysis since it was considered that free carbon materials would sediment [43].

The evaluation of cell membrane integrity was performed by staining *C. marina* cells with propidium iodide (PI, Invitrogen Life Technologies, Alfagene, Lisboa, Portugal) at 1 µg·mL^−1^. PI indicates membrane damage as it is a non-permeant red fluorescent double-charged cationic molecule that intercalates the double-stranded DNA of membrane-lesioned cells [44]. In turn, bacteria metabolic changes were assessed using 5(6)-Carboxyfluorescein diacetate (5-CFDA, Sigma–Aldrich, Taufkirchen, Germany) at 5 µg·mL^−1^ final concentration. 5-CFDA is an uncharged nonfluorescent lipophilic substrate that is hydrolyzed to fluorescent carboxyfluorescein (CF) by unspecific esterases in the cytoplasm of metabolic active cells [44]. Lastly, ROS production was evaluated by staining *C. marina* cells with 2′,7′-dichlorofluorescein diacetate (DCFH-DA, Sigma–Aldrich, Taufkirchen, Germany) at 10 µM [45]. DCFH-DA is a nonfluorescent molecule that freely penetrates the cell membrane. DCFH-DA is hydrolyzed by intracellular esterase to produce dichlorodihydrofluorescein (DCFH), which is then oxidized by ROS, producing fluorescent 2′,7′-dichlorofluorescein (DCF) [46]. Bacterial suspensions were stained with dyes for 30 min at 25 °C in absence of light. After staining, cells were analyzed in a CytoFLEX flow cytometer model V0-B3-R1 (Beckman Coulter, Brea, CA, USA) using the CytExpert software (version 2.4.0.28, Beckman Coulter, Brea, CA, USA). Bacteria were gated based on the forward (FSC) and side scatter (SSC) signals. Samples were acquired at a flow rate of 10 µL·min^−1^. Since there is an overlap between the signal of non-treated and treated cells stained with 5-CFDA and DCFH-DA, the mean intensity of fluorescence (MIF) at FL1 (fluorescent detector, 530 nm) was recorded for these dyes. As the fluorescence changes induced by the GNP exposure were clearly detected for cells stained with PI, the percentage of PI-positive (PI(+)) cells at FL3 (fluorescent detector, 610 nm) was registered.

#### 2.3.2. Epifluorescence Microscopy

Cell membrane integrity was also evaluated by staining bacteria, non-exposed and exposed to GNP 5% (*w*/*v*) (as described above), with a Live/Dead*^®^* (L/D) BacLight™ Bacterial Viability kit (Invitrogen Life Technologies, Alfagene, Portugal) as previously described [47] and performing epifluorescence microscopy analysis (Leica DM LB2, Wetzlar, Germany). In this staining, the green fluorescent dye (SYTO*^®^* 9) penetrates all cells, while red fluorescent dye (PI) penetrates only cells with compromised membranes [48].

#### 2.3.3. Scanning Electron Microscopy (SEM)

The effect of GNP on the cell membrane was further studied by SEM. For this analysis, both samples (bacteria non-exposed and exposed to GNP 5% (*w*/*v*)) were dehydrated [47] and sputter-coated with an Au/Pd thin film by using the SPI Module Sputter Coater equipment (SPI^®^ Supplies, West Chester, PA, USA). The SEM/EDS exam was performed using a high resolution (Schottky) Environmental Scanning Electron Microscope with X-ray Microanalysis and Electron Backscattered Diffraction analysis (FEI Quanta 400 FEG ESEM/EDAX Genesis X4M; FEI Company, Hillsboro, OR, USA) in high-vacuum mode at 15 kV.

### 2.4. Surface Preparation

Biofilm formation was studied using carbon-modified coatings—GNP/polydimethylsiloxane (PDMS) composites—and bare PDMS (control). Glass coupons (1 cm × 1 cm; Vidraria Lousada, Lda, Lousada, Portugal) were used as a substrate for coating, after proper cleaning and disinfection [49].

GNP/PDMS composites were produced through a bulk mixing process that entailed the incorporation of commercially available GNP aggregates (Alfa Aesar, Thermo Fisher Scientific, Erlenbachweg, Germany) into the PDMS base elastomer (Sylgard 184 Part A, Dow Corning, Midland, MI, USA viscosity = 1.1 cm^2^·s^−1^; specific density = 1.03) as described by Vagos et al. [50]. In order to assess the best GNP load to reduce bacterial adhesion, GNP were incorporated into the PDMS matrix at different loadings (1, 2, 3, 4, and 5 wt%). The mixture was agitated for 30 min at 500 rpm before being sonicated (Hielscher UP400S, at 200 W and 12 kHz) for 60 min to promote GNP dispersion. The composites were then placed in an ultrasonic bath (Selecta Ultrasons, Lisbon, Tecnilab, Portugal) for 30 min to eliminate any leftover air bubbles. The curing agent (Sylgard 184 Part B, Dow Corning) was then gently added to the base-polymer/GNP combination (in a 10:1 A:B ratio) and carefully mixed. Spin coating (Spin150 PolosTM, Caribbean, The Netherlands) for 1 min with a 500 rpm ramp to 6000 rpm was used to apply the GNP/PDMS composites as a thin layer on top of the glass coupons [51]. Finally, to induce the curing process, coated coupons were incubated overnight in an oven at 80 °C [52]. PDMS surfaces were produced by the same procedure and were used as control.

### 2.5. Surface Characterization

#### 2.5.1. Atomic Force Microscopy (AFM)

AFM studies were performed using a Veeco Multimode NanoScope IVa (Digital Instruments, Tonawanda, NY, USA) in tapping mode in air using TESP-V2 probes (Bruker, Billerica, MA, USA) with a spring constant of 26 N·m^−1^. The surface roughness was determined from five random areas (30 × 30 µm) at room temperature. Surface roughness calculations and 2D images were obtained using the Nanoscope Analysis Software from Bruker. The roughness height parameter determined was the average roughness (*R_a_*), which calculates the average absolute deviation of the roughness irregularities from the mean line over one sampling length [53].

#### 2.5.2. Scanning Electron Microscopy

To examine the morphology of the coated surfaces, SEM analyses were performed using a high resolution (Schottky) Environmental Scanning Electron Microscope with X-ray Microanalysis and Electron Backscattered Diffraction analysis (FEI Quanta 400 FEG ESEM/EDAX Genesis X4M; FEI Company, Hillsboro, OR, USA). Samples were coated for 120 sec and under a 15 mA current with a Au/Pd thin film, by sputtering, using the SPI Module Sputter Coater equipment (SPI^®^ Supplies, West Chester, PA, USA).

#### 2.5.3. Thermodynamic Analysis

Through contact angle measurements [54] and subsequent application of the van Oss approach [55], the hydrophobicity of the bacterial cells and coated surfaces was determined.

To prepare bacterial substrata, cell suspensions containing 1 × 10^9^ cells·mL^−1^ were filtered using cellulose membranes, according to the method described by Busscher et al. [56]. Contact angle measurements were performed at room temperature (23 ± 2 °C) using the sessile drop method in a contact angle meter (Dataphysics OCA 15 Plus, Filderstadt, Germany). Water, formamide, and α-bromonaphthalene (Sigma-Aldrich Co., St. Louis, MO, USA) were used as reference liquids, in three independent assays. For each assay, a minimum of 10 measurements were carried out.

According to the van Oss approach [55], a pure substance’s total surface free energy (*γ^TOT^*) results from the sum of the apolar Lifshitz–van der Waals (*γ^LW^*) and the polar Lewis acid–base (*γ^AB^*) free energy components (Equation (1)).
(1)$$\gamma^{TOT}=\gamma^{LW}+\gamma^{AB}$$

In turn, the *γ^AB^* free energy component depends on the electron acceptor (*γ^+^*) and electron donor (*γ*^−^) parameters, as shown in Equation (2).
(2)$$\gamma^{AB}=2\sqrt{\gamma^{+}\gamma^{-}}$$

By measuring the contact angles (*θ*) of a solid surface (*s*) with three distinct liquids (*l*) of known surface tension components, it is possible to determine surface free energy components. This process involves an equation of this sort for each liquid, resulting in three equations that must be solved simultaneously (Equation (3)).
(3)$$\left(1+\cos\theta \right)\gamma_{l}=2\left(\sqrt{\gamma_{s}^{LW}\gamma_{l}^{LW}}+\sqrt{\gamma_{s}^{+}\gamma_{l}^{-}}+\sqrt{\gamma_{s}^{-}\gamma_{l}^{+}}\right)\;$$

The degree of hydrophobicity of a certain solid surface is expressed as the free energy of interaction (Δ*G*, mJ·m^−2^) between two entities of that surface immersed in polar liquid (such as water (*w*), as a model solvent), according to Equation (4).
(4)$$\Delta G=-2{\left(\sqrt{\gamma_{s}^{LW}}-\sqrt{\gamma_{w}^{LW}}\right)}^{2}-4\left(\sqrt{\gamma_{s}^{+}\gamma_{w}^{-}}+\sqrt{\gamma_{s}^{-}\gamma_{w}^{+}}-\sqrt{\gamma_{s}^{+}\gamma_{s}^{-}}-\sqrt{\gamma_{w}^{+}\gamma_{w}^{-}}\right)\;\;$$

Based on this approach, if the interaction between the two entities is stronger than the interaction of each one with water (Δ*G* < 0 mJ·m^−2^), the material is deemed hydrophobic (free energy of interaction is attractive); contrarily, if Δ*G* > 0 mJ·m^−2^, the material is considered to be hydrophilic (free energy of interaction is repulsive) [57].

Regarding the interaction between a solid surface (*s*) and bacterial cells (*b*), the total interaction energy, Δ*G^Adh^*, can be determined using Equation (5):(5)$$\Delta G^{Adh}=\gamma_{sb}^{LW}-\gamma_{sw}^{LW}-\gamma_{bw}^{LW}+2\left[\sqrt{\gamma_{w}^{+}}\left(\sqrt{\gamma_{s}^{-}}+\sqrt{\gamma_{b}^{-}}-\sqrt{\gamma_{w}^{-}}\right)+\sqrt{\gamma_{w}^{-}}\left(\sqrt{\gamma_{s}^{+}}+\sqrt{\gamma_{b}^{+}-\sqrt{\gamma_{w}^{+}}}\right)-\sqrt{\gamma_{s}^{+}\gamma_{b}^{-}}-\sqrt{\gamma_{s}^{-}\gamma_{b}^{+}}\right]$$

If Δ*G^Adh^* < 0 mJ·m^−2^, the adhesion of bacteria to the material is thermodynamically favorable, whereas when Δ*G^Adh^* > 0 mJ·m^−2^, the adhesion is not favorable [58].

### 2.6. Initial Bacterial Adhesion Assay

An initial bacterial adhesion assay was performed using 12-well microtiter plates (VWR International, Carnaxide, Portugal). Two coupons of each coating material (GNP/PDMS at 1, 2, 3, 4, and 5 wt% loadings and PDMS, control) were used.

Firstly, the microplates and down-facing coated glass coupons were UV-sterilized for 1 h. Then, the coupons were fixed to the microplates’ wells with the surfaces facing upwards, using transparent double-sided adhesive tape [49]. After a second round of UV sterilization for 1 h, the wells were inoculated with 3 mL of *C. marina* bacterial suspension at OD_610_ = 0.1. The microplates were incubated at 25 °C in an orbital shaker with a 25 mm diameter (Agitorb 200ICP, Norconcessus, Ermesinde, Portugal) at 185 rpm. This shaking frequency was selected based on previous studies that reported that, for this orbital shaker, 185 rpm corresponds to an average shear rate of 40 s^−1^ [59], which, in turn, can be compared to the shear rate experienced by a ship in a harbor (50 s^−1^) [60].

Considering that, in marine environments, bacterial adhesion occurs within the first 24 h [61], the initial adhesion assay was performed after 7.5 h of incubation, according to Faria et al. [62]. At this point, the coupons were removed, and each coupon was immersed in 2 mL of sterile 8.5 mg·mL^−1^ sodium chloride solution (NaCl) and vortexed for 3 min to allow adhered bacteria to detach from the surface. Serial dilutions were performed and 100 μL of each cell suspension was then analyzed by flow cytometry (CytoFLEX V0-B3-R1, Beckman Coulter, Brea, CA, USA) in order to determine the number of cells that had adhered to each coupon.

### 2.7. Biofilm Formation Assays

Similar to the protocol described in the previous section, biofilm formation assays were performed using 12-well microtiter plates (VWR International, Carnaxide, Portugal), incubated at 25 °C in an orbital shaker with a 25 mm diameter (Agitorb 200ICP, Norconcessus, Ermesinde, Portugal) at 185 rpm (average shear rate of 40 s^−1^).

Based on the results of the initial bacterial adhesion assay, as well as on the MIC of GNP determined for *C. marina*, two surfaces, 5% wt GNP/PDMS (G5/PDMS) and PDMS, were tested. Biofilm development was monitored for 6 weeks (42 days), since this period corresponds to approximately half of the minimum economically viable interval accepted for undersea system maintenance and hull cleaning [3,59]. During this period, the VNSS culture medium was replaced twice a week. Two independent biofilm formation experiments were performed with two samples each (*n* = 4).

Preliminary studies demonstrated that seven days was the minimum time to obtain reliable and quantifiable results through the methods used in this work; thus, biofilm analysis was performed every seven days [59,63]. On each sampling day, two coupons from each independent assay for each surface were analyzed.

### 2.8. Biofilm Analysis

#### 2.8.1. Biofilm Total Cell Count

To determine the number of cells in the biofilms, each coupon was immersed in 2 mL of sterile 8.5 mg·mL^−1^ NaCl and vortexed for 3 min to allow adhered bacteria to detach from the surface [64]. Subsequently, serial dilutions were performed and the number of cells per cm^2^ was determined through flow cytometric analysis by acquiring 10 µL of cell suspension at a flow rate of 10 µL·min^−1^ (CytoFLEX V0-B3-R1, Beckman Coulter, Brea, CA, USA).

#### 2.8.2. Biofilm Thickness and Structure

Images from *C. marina* biofilms developed on different surfaces were captured and analyzed as reported by Romeu et al. [59] through OCT (Thorlabs Ganymede Spectral Domain Optical Coherence Tomography system with a central wavelength of 930 nm, Thorlabs GmbH, Dachau, Germany).

Prior to biofilm analysis, to remove any loosely adhered bacteria, the culture medium was carefully removed from the microplate wells, coupons were washed once, and wells were re-filled with 3 mL of sterile 8.5 mg·mL^−1^ NaCl.

Since biofilms are mainly composed of water [65], the refractive index was set to 1.40, close to the refractive index of water (1.33). To guarantee the accuracy and reliability of the results, 2D and 3D imaging were performed with a minimum of three fields of view for each coupon.

For image analysis, the bottom of the biofilm was identified as the best-fitting parable and hyperboloid in 2D and 3D pictures, respectively, that connected the white pixels caused by light reflection on the substratum surface. The gray-level histogram of the entire image was used to calculate a gray-level threshold that allowed us to differentiate the biofilm from the backdrop. The upper contour line of the biofilm was determined to be the image’s pixels that were connected to the base of the biofilm with a gray-level over the established threshold [66]. Any objects not connected to the bottom were thus not considered to be a part of the biofilm. Mean biofilm thickness was calculated as a function of the number of pixels between the bottom of the biofilm and the upper contour line, for each vertical line of the image [58].

In order to further assess biofilm architecture, for the last sampling day, biofilm porosity and biofilm biovolume were determined as described by Romeu et al. [31]. The percentage of biofilm porosity corresponds to the ratio between the volume of non-connected pores and the total biofilm volume, whereas biovolume pertains to the total biofilm volume per area of the region of interest.

### 2.9. Statistical Analysis

Data analysis was performed using the IBM SPSS Statistics version 28.0 for MacOS (IBM SPSS, Inc., Chicago, IL, USA). Descriptive statistics were used to calculate the mean and standard deviation (SD) for the contact angles, surface roughness, number of biofilm cells, biofilm thickness, biofilm porosity, and biofilm biovolume. Differences between the results obtained for PDMS and G5/PDMS in each of these parameters were evaluated using two-sample *t*-tests since the variables were normally distributed. Statistically significant differences were considered for *p*-values < 0.05.

## 3. Results and Discussion

In this study, the antifouling performance of a graphene-based coating was evaluated using *Cobetia marina*, a model biofilm-forming marine bacterium, under conditions mimicking those of a real marine setting. Furthermore, graphene’s mechanisms of action were investigated, to gain a further understanding of its antibacterial and antiadhesive properties.

*C. marina* cells were exposed to 5% GNP (*w*/*v*) for 24 h in order to characterize the mechanisms of action of GNP. After this period, the cells were stained with PI (a membrane integrity marker), 5-CFDA (a metabolic activity marker), and DCFH-DA (a ROS production indicator) and analyzed by flow cytometry (Figure 1).

Figure 1a,b shows the fluorescence intensity (FI) and the percentage of PI-positive cells of *C. marina* cells non-treated and treated with GNP 5% and stained with PI. Results indicated that GNP exposure caused membrane damage in about 40% of the cell population, as demonstrated by the percentage of PI(+) cells. Additionally, epifluorescence microscopy and SEM analysis further demonstrated the cell membrane damage caused by GNP exposure (Appendix A). This finding is consistent with the proposed GNP mechanism of action which postulates that the direct contact between the bacterial membranes and the sharp edges of graphene sheets or wrapping and trapping of bacterial membranes by the nanosheets induce cell membrane damage [38,51]. In addition, the analysis of non-treated *C. marina* cells versus those treated with GNP 5% and stained with 5-CFDA (Figure 1c,d) suggested that cells exposed to GNP displayed higher metabolic activity (approximately 1.5-fold) than non-treated cells (control). The changes in bacteria metabolism can be a consequence of the oxidative stress triggered by GNP exposure [67]. Bacteria can adapt to unfavorable environmental stresses through the activation of protective mechanisms. The regulation of bacterial stress responses occurs at different cellular levels, leading to changes in gene expression, protein activity, and cellular metabolism [68]. In parallel, the assessment of endogenous ROS production by staining the cells with DCFH-DA (Figure 1e,f) showed that GNP exposure led to ROS production, as demonstrated by the higher fluorescence intensity (3.6-fold) of treated cells compared to the control. It is known that ROS production depends on the physiological state of the cells, specifically on changes in metabolism as a result of stress [69]. Hence, these results suggest that oxidative stress imposed by GNP occurs through a ROS-dependent pathway, because of highly cumulated intracellular ROS [67]. This finding is also in accordance with the hypothesized mechanism of action for graphene, which involves ROS generation [38].

Since laboratory assays commonly used to test the antifouling effectiveness of marine surfaces and the dynamics of biofilm formation are laborious and time-consuming [70], a range of surfaces: PDMS (control) and GNP/PDMS composites containing different GNP loadings—1 wt% GNP/PDMS (G1/PDMS), 2 wt% GNP/PDMS (G2/PDMS), 3 wt% GNP/PDMS (G3/PDMS), 4 wt% GNP/PDMS (G4/PDMS), and 5 wt% GNP/PDMS (G5/PDMS)—were first screened for their anti-adhesion potential through the method proposed by Faria et al. [62]. According to this method, initial adhesion assays (7.5 h) can be used to estimate marine biofilm development.

Although no statistically significant differences were registered, results indicate that there is a tendency for the G5/PDMS surface (mean value of 1.97 × 10^7^ cells·cm^−2^) to induce a higher reduction in the number of adhered cells compared to bare PDMS control (mean value of 4.23 × 10^7^ cells·cm^−2^) (Appendix A). Additionally, *C. marina* showed a MIC value to GNP of 5% (*w*/*v*). Hence, based on these data, G5/PDMS composite was selected for surface characterization and long-term biofilm formation assays.

Early cell adhesion, subsequent biofilm formation, and its proliferation rely heavily on surface properties, namely wettability, topography, and roughness [71,72]. As such, the aforementioned surface characterization methods were employed to analyze the two surfaces selected for the long-term biofilm formation assays: PDMS and G5/PDMS.

Following the approach proposed by van Oss [55], the hydrophobicity of the two surfaces and the tested microorganism was determined. Contact angle measurements (*θ_l_*) and free energy of interaction (∆*G*) values are presented in Table 1. Both surfaces displayed contact angles with water higher than 90° (Table 1 and Appendix A), as well as negative free energy of interaction values (∆*G*), suggesting that they are hydrophobic. Furthermore, since the free energy of interaction values determined for each surface are very close (−63.1 mJ·m^−2^ for PDMS and −67.5 mJ·m^−2^ for G5/PDMS), it can be inferred that they display a similar degree of hydrophobicity. According to the literature, the model bacterium used in these biofilm formation assays, *C. marina*, shows a preference for adhesion on hydrophobic surfaces [17,20]. As such, biofilm formation can be expected on both surfaces. *C. marina*’s contact angle value with water (*θ_W_* = 37.4° ± 1.7°; *θ_W_* < 90°) and free energy of interaction (∆*G* = 14.9 mJ·m^−2^, ∆*G* > 0) indicate that this microorganism is hydrophilic, which is confirmed by the literature [20]. In order to predict the extent of *C. marina*’s adhesion to the tested surfaces, the free energy of adhesion (∆*G^Adh^*) values between the microorganism and each surface were determined (Table 1). The results indicate that, in theory, the adhesion of *C. marina* to both tested surfaces is thermodynamically favorable (∆*G^Adh^* < 0).

To further assess the surfaces’ properties and how they might impact biofilm formation, surface topography and roughness were determined by AFM in tapping mode (Figure 2). The analysis of surface topography revealed that G5/PDMS displayed an average roughness (*R_a_*) value about 100 times higher than that of bare PDMS.

Surface characterization was complemented by SEM analysis (Figure 3). This microscopic technique allows the evaluation of the morphological details of surfaces at nanometer resolution [73]. PDMS stood out as more homogeneous and smoother than the G5/PDMS composite, which displayed a rough appearance, with the presence of uniformly dispersed graphene agglomerates forming small elevations on the surface of the coating (Figure 3b). These results corroborate the AFM analysis, as well as those reported by Oliveira et al. [51]. As a result of the van der Waals forces and strong π–π interactions between individual GNP sheets, the dispersion of GNP is often challenging, leading to the formation of graphene agglomerates [51,74]. In these clusters, the carbon material is more exposed, promoting its direct contact with bacteria, which, in turn, potentiates graphene’s antimicrobial activity [68,75].

Following surface characterization, *C. marina* biofilm formation on PDMS and G5/PDMS was evaluated at 185 rpm (average shear rate of 40 s^−1^) for 42 days through the analysis of the number of biofilm cells and biofilm thickness (Figure 4). As expected, considering the results of the thermodynamic analysis, *C. marina* showed biofilm development on both surfaces. Overall, the results obtained for these two parameters are in accordance. Both biofilm cells and biofilm thickness displayed an increasing trend for each surface throughout the total incubation period. Concerning total biofilm cells, this increase was particularly noticeable between incubation days 21 (on average, 2.46 × 10^7^ cells·cm^−2^ for PDMS and 1.80 × 10^7^ cells·cm^−2^ for G5/PDMS) and 28 (on average, 1.11 × 10^8^ cells·cm^−2^ for PDMS and 8.20 × 10^7^ cells·cm^−2^ for G5/PDMS). As for biofilm thickness, the most noticeable growth spurt was between incubation days 28 (on average, 46 µm for PDMS and 34 µm for G5/PDMS) and 35 (on average, 113 µm for PDMS and 70 µm for G5/PDMS). These increments may indicate that *C. marina* biofilm maturation occurred between the 3- and 5-week marks. More importantly, from incubation days 14 to 42, G5/PDMS showed consistently lower biofilm cell count and thickness values than bare PDMS (approximately 22% reduction on incubation day 14; 26% reduction on incubation day 28; 38% reduction on incubation day 35; 43% reduction on incubation day 42 for both analyzed parameters; *p* < 0.05). For the last four incubation weeks, the G5/PDMS reduction percentages were similar for both tested parameters, indicating that the GNP surface was not only able to effectively reduce the total cells but also the thickness of biofilms formed by *C. marina*, which is of extreme importance, given the role of biofilm control on marine biofouling.

Altogether, these results indicate that the graphene-based polymeric coating showed significant antibacterial and antibiofilm performance. This might be a result of the aforementioned GNP agglomerates, which trigger membrane damage and oxidative stress in *C. marina* cells, as demonstrated by the flow cytometric analysis. In fact, GNP clusters are likely to promote the membrane-piercing effect of exposed graphene particles, consequently affecting cell integrity [37,67]. It is hypothesized that the first layer of adhered cells is particularly affected by this dart-like effect, impairing the subsequent adhesion of cell layers, and, therefore, curbing long-term biofilm formation. Overall, these results corroborate the previously reported antibacterial and antifouling properties of graphene materials [32,51,75,76].

Figure 5 includes representative 3D images retrieved through OCT from biofilms formed by *C. marina* on PDMS (Figure 5a,b) and G5/PDMS (Figure 5c,d) at incubation days 21 (Figure 5a,c) and 42 (Figure 5b,d). More than the evolution of their thickness between the two incubation days, which is in accordance with the quantitative biofilm thickness results (Figure 5b), these representative 3D images allow us to observe the spatial distribution of the biofilm structures across the tested surfaces.

Moreover, the biovolume of biofilms formed on both surfaces was calculated for incubation day 42 (Figure 6a). This parameter provides an estimate of the biomass in the biofilm (µm^3^) per area of the region of interest (mm^2^) [31]. Results showed that the biofilms formed on G5/PDMS had significantly lower biovolume than those developed on the control surface (on average, 1.80 × 10^8^ µm^3^·mm^−2^ for PDMS versus 8.89 × 10^7^ µm^3^·mm^−2^ for G5/PDMS; *p* ≤ 0.01). These results corroborate the ones obtained for both biofilm total cells and biofilm thickness (Figure 4). Biofilm porosity values were also determined for both surfaces on incubation day 42 (Figure 6b). Results showed significantly lower biofilm porosity on G5/PDMS in comparison to the control surface (on average, 61% porosity on PDMS versus 27% on G5/PDMS; *p* ≤ 0.001). Figure 6c,d consists of representative 2D cross-sectional images acquired through OCT of biofilms developed on PDMS (Figure 6c) and G5/PDMS (Figure 6d) on incubation day 42. The comparison of these two images allows us to visually assess the differences in structure and porosity between biofilms formed on each tested surface. It is possible to observe that biofilms formed on G5/PDMS possess fewer empty spaces than those formed on bare PDMS, which corroborates the quantitative porosity percentage results and is in accordance with a previous study performed with GNP-based surfaces using cyanobacteria [32]. In biofilms, empty spaces, such as pores and channels, can be beneficial to ensure that nutrients and oxygen reach the cells, as well as to dilute waste products or antimicrobials [77]. Hence, biofilms with a higher percentage of empty spaces, such as those developed on the PDMS surfaces, have an advantage in terms of mass transfer and storage, and greater potential to grow when compared with less porous biofilms [78], such as the ones formed on the G5/PDMS surface.

All in all, this work demonstrates the antibiofilm potential of GNP/PDMS surfaces against *C. marina* biofilm development. The results obtained are promising, especially considering that the incorporation of pristine graphene into polymeric marine antifouling coatings is poorly documented [30]. Furthermore, the cytometric assessment of the effects of GNP on *C. marina* cells can contribute to a better understanding of graphene’s antibacterial mechanisms of action.

## 4. Conclusions

In this study, the long-term antifouling performance of GNP-based surfaces for inhibiting *C. marina* biofilm development was demonstrated. In fact, GNP/PDMS coatings, due to their surface properties, allied with GNP antimicrobial activity impacted not only biofilm composition but also its structure. Biofilms developed on the GNP composite displayed significantly reduced cell count, thickness, biovolume, and porosity in the maturation stage when compared to the control surface (bare PDMS). In addition, the comprehensive analysis of graphene’s mechanisms of action carried out in this study showed that these carbon nanomaterials cause bacterial membrane damage and induce oxidative stress mediated by ROS production, leading to cell inactivation. Overall, these results demonstrated the potential of incorporating GNP in marine paints to mitigate marine biofouling and its negative consequences.

## Figures and Tables

**Figure 1 nanomaterials-13-00381-f001:**
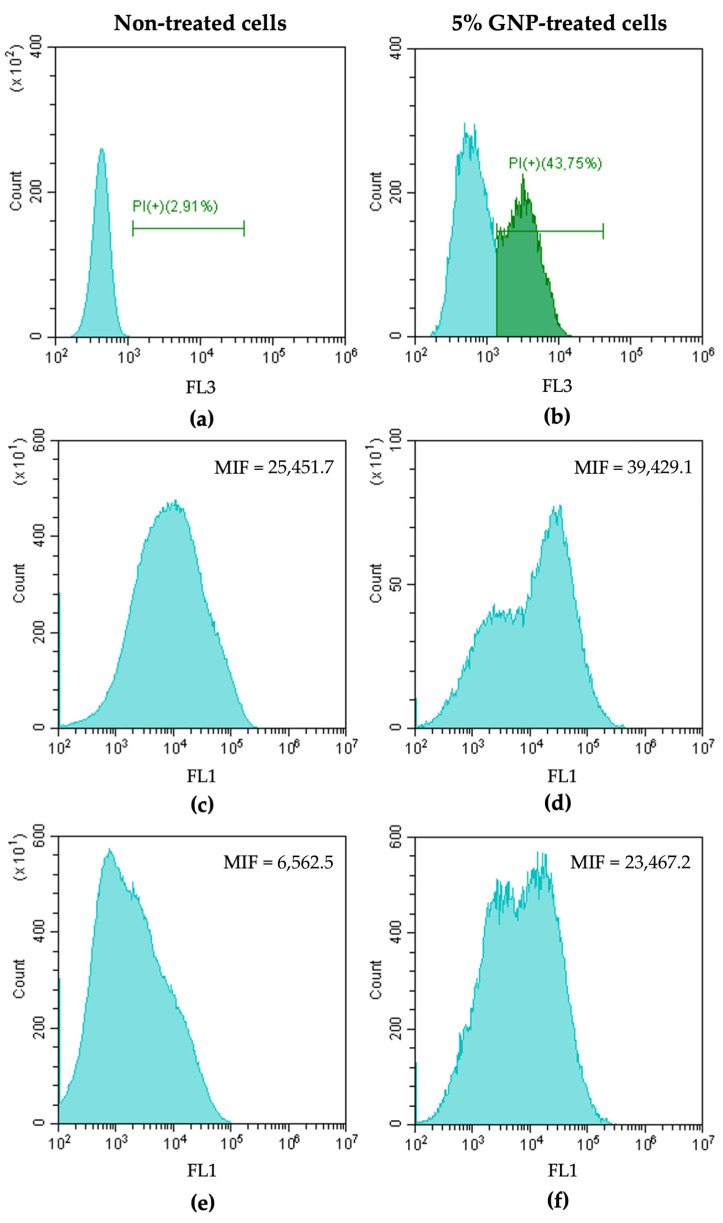
Representative flow cytometric histograms obtained for non-treated cells and cells treated with GNP 5% (*w*/*v*), stained with PI (**a**,**b**), 5-CFDA (**c**,**d**), and DCFH-DA (**e**,**f**), respectively. Results were presented as the percentage of PI-positive cells (PI(+)) and the mean intensity of fluorescence (MIF) for cells stained with 5-CFDA and DCFH-DA.

**Figure 2 nanomaterials-13-00381-f002:**
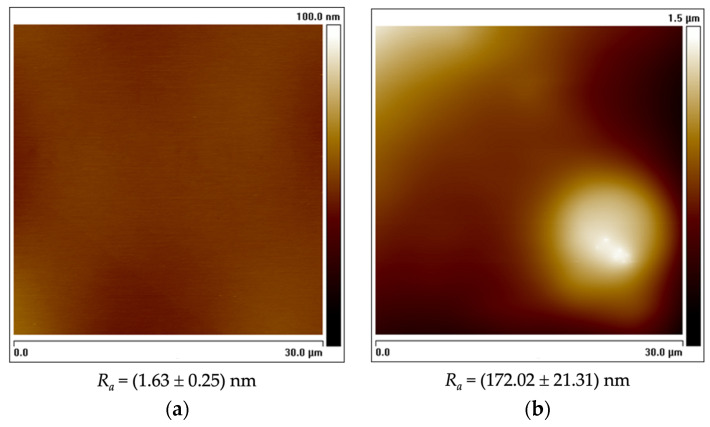
Two-dimensional AFM images of PDMS (**a**) and G5/PDMS (**b**) with a scan range of 30 × 30 µm (tapping mode). Average roughness (*R_a_*) values for each surface presented as mean ± SD. The vertical color bars correspond to the z-range (surface height range) of the respective image; they are not on the same scale to allow the visualization of surface features.

**Figure 3 nanomaterials-13-00381-f003:**
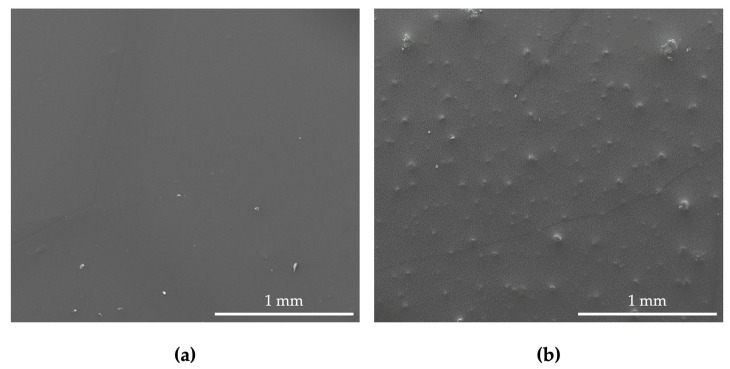
SEM images of PDMS (**a**) and G5/PDMS (**b**) (magnification of 100× and scale bars of 1 mm).

**Figure 4 nanomaterials-13-00381-f004:**
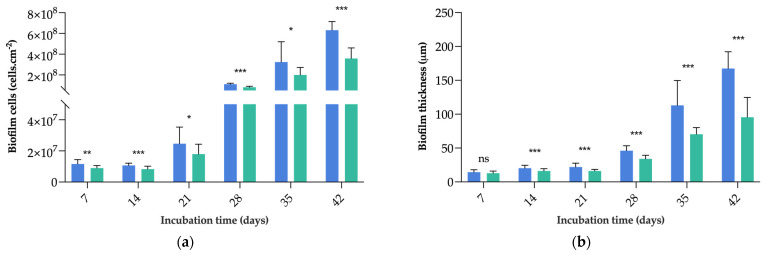
Evaluation of the *C. marina* biofilm development on PDMS (

) and G5/PDMS (

) throughout 42 days. The analyzed parameters refer to the number of biofilm cells (**a**) and biofilm thickness (**b**). Results are presented as mean ± SD. ns—*p* > 0.05; *—*p* ≤ 0.05; **—*p* ≤ 0.01; ***—*p* ≤ 0.001.

**Figure 5 nanomaterials-13-00381-f005:**
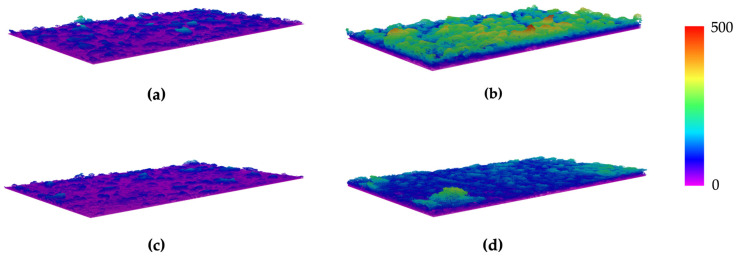
Three-dimensional OCT images of *C. marina* DSM 4741 biofilms formed on PDMS (**a**,**b**) and G5/PDMS (**c**,**d**) acquired on incubation days 21 (**a**,**c**) and 42 (**b**,**d**). Scale bar in μm.

**Figure 6 nanomaterials-13-00381-f006:**
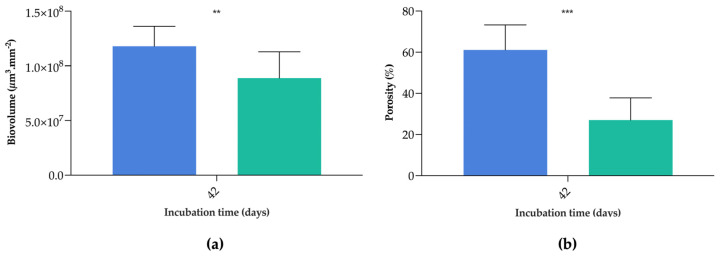
(**a**) Biovolume of biofilms formed on PDMS (

) and G5/PDMS (

), determined at incubation day 42. **—*p* ≤ 0.01. (**b**) Porosity of biofilms formed on PDMS (

) and G5/PDMS (

) determined on incubation day 42. ***—*p* ≤ 0.001. (**c**,**d**) Two-dimensional cross-sectional OCT images of *C. marina* DSM 4741 biofilms formed on PDMS (**c**) and G5/PDMS (**d**) acquired on incubation day 42. Empty spaces highlighted in light blue. White scale bars = 100 μm.

**Table 1 nanomaterials-13-00381-t001:** Contact angles with the three reference liquids (water, formamide, and α-bromonaphthalene), surface and bacteria hydrophobicity, and free energy of *C. marina*–surface interaction. Results are presented as mean ± SD.

Sample	Contact Angle (°)			Hydrophobicity (mJ·m^−2^)	*C. marina*–Surface Interaction (mJ·m^−2^)
	*θ_W_*	*θ_F_*	*θ_B_*	Δ*G*	Δ*G*^Adh^
**Surface**					
PDMS	114.3 ± 1.4	111.8 ± 1.6	88.3 ± 3.8	−63.1	−5.5
G5/PDMS	115.8 ± 2.1	112.2 ± 1.7	89.8 ± 3.7	−67.5	−6.7
**Bacteria**					
*C. marina* DSM 4741	37.4 ± 1.7	31.5 ± 2.3	55.0 ± 1.9	14.9	n.a.

Abbreviations: *θ_W_*—contact angle with water; *θ_F_*—contact angle with formamide; *θ_B_*—contact angle with α-bromonaphthalene; ∆*G*—free energy of interaction; ∆*G^Adh^*—free energy of adhesion between *C. marina* and the tested surfaces; n.a.—not applicable.

## Data Availability

The data presented in this study are available from the corresponding author upon request.

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
