# Peer review of "Graphene-Based Coating to Mitigate Biofilm Development in Marine Environments"

_nanomaterials, 2023, doi:10.3390/nano13030381_

Round 1

Reviewer 1 Report

This work reported by Sousa-Cardoso et al. assessed the effect of a graphene-based coating on biofilm development by C. marina over a long-term in vitro assay performed under hydrodynamic conditions mimicking a real marine setting. Meanwhile, the authors investigated the mechanism of action of pristine graphene nanoplatelets. The experiments were well-designed with an interesting presentation of results and discussion. Therefore, I am of the opinion that this work could be acceptable for publication in the journal of Nanomaterials. However, additional experimental results are required in the revision to answer some missing points. In addition, the following suggestions are expected to make the discussions more accurate and clearer.

(1) The authors mentioned that GNP exposure caused membrane damage to C. marina. However, the analysis results by flow cytometry cannot provide proof. Please add relevant experiments and data to clarify this result such as SEM of dead cells.

(2) The zeta-potential of materials is recommended for investigation. Positively charged materials are known to damage the cell membranes of bacteria.

(3) It is suggested that the authors should unify the writing format of full text, such as 8.5 mg·mL-1 in Line 251 and 10 μL/min in Line 254.

(4) Please check and unify the format of the reference.

(5) It is suggested that the authors polish the language of the manuscript.

(6) The authors may consider newest references. A few suggestion: a. Packaging and degradability properties of polyvinyl alcohol/gelatin nanocomposite films filled water hyacinth cellulose nanocrystals; b. Plastic crisis underscores need for alternative sustainable-renewable materials; c. Effects of chitin nanocrystals on coverage of coating layers and water retention of coating color; d. Cellulose-based thermosensitive supramolecular hydrogel for phenol removal from polluted water; e. Volatile organic compounds in water matrices: Recent progress, challenges, and perspective; f. Production of magnetic sodium alginate polyelectrolyte nanospheres for lead ions removal from wastewater.

Reviewer 3 Report

In this work, Sousa-Cardoso et al. have investigated the effect of pristine graphene nanoplatelet coating on the biofilm development by marine bacteria Cobetia marina. Their results demonstrate that GNP coated surface can inhibit the biofilm development significantly, making GNP a promising material in the application of marine antifouling coating. This work is of interest to the audience of Nanomaterials.  There are few concerns, though, that need to be addressed before acceptance.

1.     This work seems to be an extended study following their previous work on the effect of GNP coating on the biofilm development by marine cyanobacterial (Coatings 202212, 1775). The authors need to provide a clear justification in the introduction why a different marine bacteria Cobetia marina deserves a separated study. Additionally, the authors may want to compare the biofilms formed by these two kinds of bacteria.

2.     What is the rational that two different representations are used in Figure 1? In other words, why the PI stained cells are characterized by PI-positive ratio, whereas the 5-CFDA and DCFH-DA stained cells are characterized by MIF?  The GNP-treated cells exhibit two peaks in the fluorescence intensity profiles, whereas the non-treated cells only show one peak. The authors are recommended to provide additional interpretation on this observation.

Round 2

Reviewer 1 Report

The authors have made good revision.